# Hybrid predictive coding: Inferring, fast and slow

**Alexander Tscshantz**[1,2,3☯]*, **Beren Millidge** [1,2,4☯]*, **Anil K. Seth**[1,3], **Christopher L. Buckley**[1,2]

**1** Sussex AI Group, Department of Informatics, University of Sussex, Brighton, United Kingdom, **2** VERSES Research Lab, Los Angeles, California, United States of America, **3** Sussex Centre for Consciousness Science, University of Sussex, Brighton, United Kingdom, **4** Brain Networks Dynamics Unit, University of Oxford, Oxford, United Kingdom

☯ These authors contributed equally to this work.
* tschantz.alec@gmail.com (AT); beren@millidge.name (BM)

**Data Availability Statement:** All data are in the manuscript and/or supporting information files. Our code is available on github online at https://github.com/alec-tschantz/pybrid.

**Funding:** This work was supported by the European Research Council (Advanced Investigator

## Abstract

Predictive coding is an influential model of cortical neural activity. It proposes that perceptual beliefs are furnished by sequentially minimising "prediction errors"—the differences between predicted and observed data. Implicit in this proposal is the idea that successful perception requires multiple cycles of neural activity. This is at odds with evidence that several aspects of visual perception—including complex forms of object recognition—arise from an initial "feedforward sweep" that occurs on fast timescales which preclude substantial recurrent activity. Here, we propose that the feedforward sweep can be understood as performing *amortized* inference (applying a learned function that maps directly from data to beliefs) and recurrent processing can be understood as performing *iterative* inference (sequentially updating neural activity in order to improve the accuracy of beliefs). We propose a *hybrid predictive coding* network that combines both iterative and amortized inference in a principled manner by describing both in terms of a dual optimization of a single objective function. We show that the resulting scheme can be implemented in a biologically plausible neural architecture that approximates Bayesian inference utilising local Hebbian update rules. We demonstrate that our hybrid predictive coding model combines the benefits of both amortized and iterative inference—obtaining rapid and computationally cheap perceptual inference for familiar data while maintaining the context-sensitivity, precision, and sample efficiency of iterative inference schemes. Moreover, we show how our model is inherently sensitive to its uncertainty and adaptively balances iterative and amortized inference to obtain accurate beliefs using minimum computational expense. Hybrid predictive coding offers a new perspective on the functional relevance of the feedforward and recurrent activity observed during visual perception and offers novel insights into distinct aspects of visual phenomenology.

## Author summary

Predictive Coding Networks (PCNs) are a neurobiologically plausible model of cortical processing that can be applied to machine learning tasks. However, they require a

Grant 101012954 to AKS), the Canadian Institute
for Advanced Research (Program in Brain, Mind,
and Consciousness to AKS), the Dr. Mortimer and
Theresa Sackler Foundation (part PhD studentship
to AT via the Centre for Consciousness Science),
the School of Engineering and Informatics at the
University of Sussex (part PhD studentship to AT).
It was additionally supported by the Biotechnology
and Biological Sciences Research Council (Grant
BB/P022197/1 to CLB) and (Grant BB/S006338/1
to BM). The funders had no role in study design,
data collection and analysis, decision to publish, or
preparation of the manuscript.

**Competing interests:** The authors have declared
that no competing interests exist.

computationally costly inference phase to generate predictions. We propose adding an
amortized feedforward network to the model which learns to predict the outcome of itera-
tive inference, and uses these predictions to initialize the predictive coding network—an
approach we call hybrid predictive coding. This allows our hybrid model to perform
simultaneous classification and generation and can be trained much faster and with less
data than a standard PCN. Our model can also naturally and adaptively vary its computa-
tion time according to task demands, and may also help shed light on the neurocomputa-
tional basis of some otherwise difficult-to-understand aspects of visual phenomenology,
thus suggesting that the brain may utilize a similar hybrid inference approach in visual
processing.

# 1 Introduction

A classical view of perception is as a primarily bottom-up pipeline, whereby signals are pro-
cessed in a feed-forward manner from low-level sensory inputs to high-level conceptual repre-
sentations [1–3]. In apparent contrast with this classical bottom-up view, a family of
influential theories—originating with von Helmholtz in the 19th Century—have cast percep-
tion as a process of (approximate) Bayesian inference, in which prior expectations are com-
bined with incoming sensory data to form perceptual representations [4–8]. Under this
Bayesian perspective, the loci of perceptual content reside predominantly in top-down predic-
tions rather than in the sequential refinement of bottom-up sensory data.

In visual perception, top-down signalling has long been recognised as playing several
important functional roles [9]—for instance, in attentional modulation [10], in the goal-
directed shaping of stimulus selection [11, 12], and in establishing recurrent loops that have
been associated with conscious experience [13]. At the same time, bottom-up signalling has
been convincingly linked to rapid perceptual phenomena such as gist perception and context-
independent object recognition [9, 14–17]. These and other disparate findings have fueled a
long-standing debate over the respective contributions of bottom-up and top-down signals to
visual perceptual content [9, 18–21]. Although classical bottom-up perspectives are often con-
trasted with Bayesian top-down theories in this debate, more nuanced pictures have also been
proposed in which bottom-up and top-down signals both contribute to perceptual content,
but in distinct ways [21–23]. Such proposals, however, have largely remained conceptual.
Here, we develop, and illustrate with simulations, a novel computational architecture in which
top-down and bottom-up signalling is adaptively combined to bring about perceptual repre-
sentations within an extended predictive coding paradigm. We call this architecture *hybrid
predictive coding* (HPC). We show that while both bottom-up and top-down signals convey
predictions about perceptual beliefs, they implement different approaches to inference (amor-
tized and iterative inference, respectively). Our model retains the benefits of both approaches
to inference in a principled manner, and helps explain several empirical observations that
have, until now, evaded a parsimonious explanation in terms of Bayesian inference.

Predictive coding is a highly influential framework in theoretical neuroscience which origi-
nated in signal processing theory and proposes that top-down signals in perceptual hierarchies
convey predictions about the causes of sensory data [6, 8, 24–32]. Predictive coding is usually
considered in systems with multiple hierarchical layers [30, 33–35], where each layer learns to
predict (or generate) the activity of the layer below it (with the lowest layer predicting sensory
data). In this setting, bottom-up signals convey prediction errors—the difference between pre-
dictions and data—whereas top-down signals convey the predictions (although see alternative

predictive coding schemes which depart from this template [36], although in different ways to our proposed hybrid predictive coding model.). By minimising prediction errors, a system can both learn a generative model of its sensory data and infer the most likely causes of that data in a hierarchical fashion [37]. The resulting scheme can be interpreted as performing variational inference [29], an optimisation procedure that approximates Bayesian inference [38–40].

Predictive coding can account for a wide range of neurophysiological evidence and provides a compelling account of top-down signals in visual perception [41]. However, to extract meaningful representations from sensory data, predictive coding must iteratively minimise prediction errors over multiple sequential steps [42, 43]. We refer to such procedures as *iterative* inference, as they require multiple iterations to furnish perceptual beliefs [44–46]. In neural terms, this would imply that multiple cycles of recurrent activity are required to perceive a stimulus [17, 33, 47]. However, empirical studies have consistently demonstrated that many aspects of human visual perception can occur remarkably rapidly, often within 150ms of stimulus onset [14, 48–50]. This evidence of rapid perception—such as in gist perception or context-free object recognition—is difficult to reconcile with a computational process that requires several sequential steps to arrive at perceptual representations.

In machine learning, *amortised* inference provides an elegant alternative to iterative inference [51, 52] which is well suited to rapid processing. Rather than iteratively updating beliefs for each stimulus, amortised inference learns a parameterised function (such as a neural network) that maps directly from the data to (the parameters of) beliefs. The parameters of this function are optimised across the available dataset, and once learned, inference proceeds simply by applying the learned function to new data. Thus, amortised inference provides a plausible mechanism for extracting beliefs rapidly and efficiently via feed-forward, bottom-up processing [47, 53].

Our novel neural architecture—*hybrid predictive coding* (HPC)—combines (standard) iterative predictive coding with amortised inference, so that *both* bottom-up *and* top-down signals convey predictions (and where prediction errors also flow in both directions) (see Fig 1). The architecture comprises several hierarchical layers, with top-down signals predicting the activity of the subordinate layer and bottom-up signals predicting the superordinate layer's activity. As with predictive coding, top-down predictions learn to generate the data hierarchically,

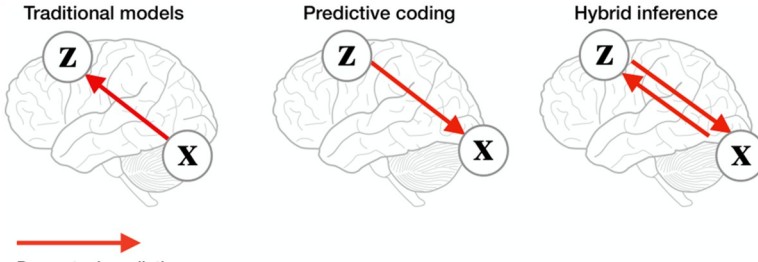

**Fig 1. Bottom-up and top-down perception.** One classical view of perception is as a primarily bottom-up process, where sensory data **x** is transformed into perceptual representations **z** through a cascade of feedforward feature detectors. In contrast, predictive coding suggests that the brain solves perception by modelling how perceptual representations **z** generate sensory data **x**, which is a fundamentally top-down process. Although there is bottom-up information flow [in PC] in terms of errors, this bottom-up information conveys errors (transformed by the synaptic weights). However, in our hybrid predictive coding models, bottom-up information conveys both predictions and errors (along with top-down information). In HPC, sensory data **x** predicts perceptual representations at fast, amortized time scales, and perceptual representations **z** predict sensory data **x** at slow, iterative time scales. Our "fast and slow" model casts this integration of bottom-up and top-down signals in a common framework, allowing derivation of a testable process theory.

implementing a 'generative model' of the data. The model augments standard predictive coding by also implementing bottom-up predictions which learn to generate (the parameters of) *beliefs* at higher layers. Crucially, these bottom-up predictions learn to generate beliefs that have been optimised by iterative inference, i.e. they learn to generate approximately posterior beliefs. In this way, our model casts bottom-up processing as *amortised* inference and top-down processing as *iterative* inference. This means that both bottom-up and top-down signals convey transformations of errors and transformations of activity. This contrasts with traditional predictive coding, where bottom-up signals convey a transformation of errors while top-down signs convey a transformation of activity (predictions).

At stimulus onset, bottom-up predictions rapidly provide an initial "best guess" at perceptual beliefs, which are then refined by minimising prediction errors iteratively in a top-down fashion. Both the bottom-up and top-down processes operate using the same set of biologically plausible Hebbian learning rules, and all layers of the network operate in unison to infer a single set of consistent beliefs. Altogether, the model offers a unified inference algorithm that inherits the rapid processing of amortised inference while maintaining the flexibility, robustness, and context sensitivity of iterative inference [8, 46, 51, 54]. Hybrid predictive coding provides a method for performing inference that is both *fast* (amortized) and *slow* (iterative) [55].

The remainder of the paper is structured as follows. Section 2 provides an overview of iterative and amortised inference and describes the hybrid predictive coding (HPC) architecture. In Section 3, we present results from a series of experiments that explore several aspects of our model. Specifically, we demonstrate **a**) that HPC performs supervised and unsupervised learning simultaneously, **b**) that bottom-up, amortised predictions reduce the number of iterations required to achieve accurate perceptual beliefs, and that the trade-off between amortised and iterative inference is adaptively modulated by uncertainty, and **c**) that the generative component of the model enables learning with a limited amount of data. Together, these results show the benefits of inferential process theories that incorporate bottom-up predictions which convey perceptual content rather than just errors. Conversely, they demonstrate that bottom-up approaches to perception should benefit from incorporating top-down generative feedback. Finally, we argue that our model provides a powerful computational framework for interpreting the contributions of bottom-up and top-down signalling in terms of different aspects of visual perception.

## 1.1 Related work

Many works in machine learning have considered the notion of iterative and amortized (variational) inference [40, 51, 56], with recent work combining iterative and amortized inference in the context of perception [46] and control [44, 45, 57, 58]. This idea of combining model-free and model-based planning methods (interpreted as iterative and amortized inference) in reinforcement learning also has a long history [59, 60]. Moreover, in perception, [61] also consider a feedforward amortized sweep to initialize an iterative inference algorithm in the context of contrastive Hebbian learning algorithms which are another proposed family of biologically-plausible learning algorithms which could potentially be implemented in neural circuitry [62, 63]. Contrastive Hebbian methods differ from predictive coding in that they require both a 'free phase' where the network output is unclamped and a 'fixed phase' where the network output is clamped and then the weight update is proportional to the difference between the two phases. In a similar approach to ours, [61] show that the fixed and free phase equilibria can be amortized and predicted in a feedforward pass and that this reduces the number of inference iterations required. However, to our knowledge, our approach is the first to combine iterative and amortized inference within a predictive coding architecture, and the resulting network has many favourable theoretical properties such as requiring only local Hebbian updates and

that all dynamics and weight changes can be derived from a joint optimization on a unified energy function. In addition, given that predictive coding has been proposed as biological process theory of perception [8, 41, 42, 64], and as a way to explain the phenomenology of perceptual experience in terms of neural mechanisms [65, 66], our novel architecture also offers insights into why gist perception and focal perception have the characteristic phenomenological properties that they do. Our work is closely related to symmetric predictive estimators (SPE) [67], a model that learns separate feedforward and feedback weights using local learning rules. Similar to our work, these feedforward and feedback weights learn bottom-up and top-down projections of activity and are learned in a biologically plausible manner using local error units. Our work differs in several ways, both in technical content and theoretical motivation. In terms of technical content, the update equations have the neuron's states continually updated based on bottom-up and top-down information, rather than the temporal asymmetry found in HPC. These differences mean that the SPE does not enable the interplay of bottom-up and top-down signals of HPC. On a theoretical level, the SPE is not directly motivated by the Bayesian brain hypothesis and does not look to provide a common framework for feedforward and recurrent activity in the brain.

## 2 Methods

### 2.1 Approximate Bayesian inference

**Bayesian inference.**   To support adaptive behaviour, the brain must overcome the ambiguous relationship between sensory data and their underlying (hidden) causes in the world. For example, suppose an object reflects some pattern of light onto the retina, the brain must recover this object's identity, despite the fact that the sensory data is inherently noisy, and that multiple objects could have caused the same pattern of retinal stimulation. Such considerations have motivated the popular view that the brain uses a version of Bayesian inference [7, 68], which describes the process of forming probabilistic beliefs about the causes of data, to accomplish perception.

Formally, we can denote sensory data (e.g., the pattern of retinal stimulation) as $\mathbf{x}$ and the hidden cause of this data (e.g., the object causing the retinal stimulation) as $\mathbf{z}$. Rather than directly calculating the most likely hidden cause, a Bayesian perspective would propose that the brain infers a conditional distribution over possible causes $p(\mathbf{z}|\mathbf{x})$, referred to as the *posterior* distribution. Bayesian inference then prescribes a method for updating the posterior distribution in light of new sensory data:

$$p(\mathbf{z}|\mathbf{x}) = \frac{p(\mathbf{x}|\mathbf{z})p(\mathbf{z})}{p(\mathbf{x})} \tag{1}$$

where $p(\mathbf{x}|\mathbf{z})$ is referred to as the likelihood distribution, describing the probabilistic relationship between hidden causes and sensory data, $p(\mathbf{z})$ is the prior distribution, describing the prior probability of hidden causes, and $p(\mathbf{x}) = \int p(\mathbf{x}|\mathbf{z})p(\mathbf{z})d\mathbf{z}$ is the evidence, describing the probability of some sensory data averaged over all possible hidden causes. Bayesian inference prescribes a normative and mathematically optimal method for updating beliefs when faced with uncertainty and provides a principled approach for integrating prior knowledge and data into inferences about the world [69–71].

**Variational inference.**   While Bayesian inference provides an elegant framework for describing perception, the computations it entails are generally mathematically intractable [38]. Therefore, it has been suggested that the brain may implement approximations to Bayesian inference. In particular, it has been suggested that the brain utilises *variational inference* [38–40, 56, 72–74], which converts the intractable inference problem into a tractable

optimisation problem. Variational inference posits the existence of an *approximate posterior* $q_\lambda(\mathbf{z})$ with parameters $\lambda$, which serves as an approximation to the true posterior distribution $p(\mathbf{z}|\mathbf{x})$. The goal of variational inference is then to minimise the difference between the true and approximate posteriors, with the difference being quantified in terms of the KL-divergence $D_{\mathrm{KL}}$. The KL-divergence is an asymmetric measure of dissimilarity between two probability distributions.:

$$D_{\mathrm{KL}}[q_\lambda(\mathbf{z})\|p(\mathbf{z}|\mathbf{x})] = \mathbb{E}_{q_\lambda(\mathbf{z})}[\ln q_\lambda(\mathbf{z}) - \ln p(\mathbf{z}|\mathbf{x})] \tag{2}$$

However, to minimise Eq 2, it is still necessary to evaluate the true posterior distribution $p(\mathbf{z}|\mathbf{x})$. Variational inference circumvents this issue by instead minimising an upper bound on Eq 2, i.e., a quantity which is always greater than or equal to the quantity of interest. In particular, it minimises the *variational free energy* $\mathcal{F}$:

$$\begin{aligned} \mathcal{F}(\mathbf{z}, \mathbf{x}) &= \mathbb{E}_{q_\lambda(\mathbf{z})}[\ln q_\lambda(\mathbf{z}) - \ln p(\mathbf{z}, \mathbf{x})] \\ &= \mathbb{E}_{q_\lambda(\mathbf{z})}[\ln q_\lambda(\mathbf{z}) - \ln p(\mathbf{z}|\mathbf{x})] - \ln p(\mathbf{x}) \\ &\geq D_{\mathrm{KL}}[q_\lambda(\mathbf{z})\|p(\mathbf{z}|\mathbf{x})] \end{aligned} \tag{3}$$

Minimising variational free energy $\mathcal{F}$ will ensure that the $q_\lambda(\mathbf{z})$ tends towards an approximation of the true posterior (see the final line of Eq 3), thus implementing an approximate form of Bayesian inference. This minimisation takes place with respect to the parameters of the approximate posterior $\lambda$, and can be achieved through methods such as gradient descent.

**Learning.** Eq 3 introduces an additional joint distribution over hidden causes and sensory data $p(\mathbf{z}, \mathbf{x})$, which is referred to as the *generative model* and is expressed in terms of a likelihood and a prior $p(\mathbf{z}, \mathbf{x}) = p(\mathbf{x}|\mathbf{z})p(\mathbf{z})$. It is common to parameterise the generative model with a set of parameters $\theta$, e.g $p_\theta(\mathbf{z}, \mathbf{x})$. These parameters can then be optimised (over a slower timescale) with respect to variational free energy, thereby providing a tractable method for *learning* [39, 75, 76]. Intuitively, this is because the variational free energy provides a bound on the marginal-likelihood of observations $p_\theta(\mathbf{x})$, such that minimising free energy with respect to $\theta$ will maximise $p_\theta(\mathbf{x})$ [77]. The subscript $\theta$ highlights that the marginal likelihood is evaluated under the generative model. Minimising $\mathcal{F}$ with respect to $\theta$ will thus cause the generative model to distill statistical contingencies from the data, and by doing so, encode information about the environment. In summary, variational inference provides a method for implementing both inference and learning using a single objective—the minimisation of variational free energy.

**Iterative inference.** Variational inference provides a general scheme for approximating Bayesian inference. In practice, it is necessary to specify the approximate posterior and generative model, as well as the optimisation scheme for minimising variational free energy. A standard method is to optimise the parameters of the variational distribution $\lambda$ for each data point. Given some data point $\mathbf{x}$, we look to solve the following optimisation procedure:

$$\lambda^* = \arg\min_\lambda \mathbb{E}_{q_\lambda(\mathbf{z})}[\ln q_\lambda(\mathbf{z}) - \ln p_\theta(\mathbf{z}, \mathbf{x})] \tag{4}$$

Generally, this is achieved using iterative procedures such as gradient descent. Therefore, we refer to this mode of optimisation as *iterative* inference [44, 46], as it requires multiple iterations to converge, and the optimisation is performed for each data point individually. Heuristically, for each new data point $\mathbf{x}$, iterative inference randomly initialises $\lambda$, and then uses standard optimization algorithms such as gradient descent to iteratively minimise Eq 4. This method underwrites a number of popular inference methods, such as stochastic variational inference [78] and black box variational inference [79], and can be considered to be the

'classical' approach to variational inference. In what follows, we specify how predictive coding can be considered as a form of iterative inference.

## 2.2 Predictive coding

The predictive coding algorithm [6, 8] operates on a hierarchy of layers, where each layer tries to predict the activity of the layer below it (with the lowest layer predicting the sensory data). These predictions are iteratively refined by minimising the prediction errors, (i.e., the difference between predictions and the actual activity) of each layer. In predictive coding, the approximate posterior is defined to be a Gaussian distribution:

$$q_\lambda(\mathbf{z}) = \mathcal{N}(\mathbf{z}; \mu, \sigma^2) \tag{5}$$

where $\lambda = \{\mu, \sigma^2\}$. In a similar fashion, we assume the factors of the generative model $p_\theta(\mathbf{z}, \mathbf{x}) = p(\mathbf{x}|\mathbf{z})p(\mathbf{z})$ to also be Gaussian:

$$
\begin{aligned}
p(\mathbf{z}) &= \mathcal{N}(\mathbf{z}; \bar{\mu}, \sigma_p^2) \\
p(\mathbf{x}|\mathbf{z}) &= \mathcal{N}(\mathbf{x}; f_\theta(\mathbf{z}), \sigma_l^2)
\end{aligned}
\tag{6}
$$

where $f_\theta(\cdot)$ is some non-linear function with parameters $\theta$, and $\bar{\mu}$ is the mean of the prior distribution $p(\mathbf{z})$. In hierarchical models, $\bar{\mu}$ would not be fixed but would instead act as an empirical prior. Given these assumptions, we can rewrite variational free energy $\mathcal{F}$ as ([30]):

$$
\begin{aligned}
\mathcal{F}(\mu, \mathbf{x}) = \frac{1}{2\sigma_l} \quad &\boldsymbol{\varepsilon}_l^2 + \frac{1}{2\sigma_p} \boldsymbol{\varepsilon}_p^2 + \frac{1}{2} \ln\left(\sigma_l \sigma_p\right) \\
&\boldsymbol{\varepsilon}_l = \mathbf{x} - f_\theta(\mu) \\
&\boldsymbol{\varepsilon}_p = \mu - \bar{\mu}
\end{aligned}
\tag{7}
$$

where $\boldsymbol{\varepsilon}_l$ and $\boldsymbol{\varepsilon}_p$ are the *prediction errors*. The term $f_\theta(\mu)$ can be construed as a *prediction* about the sensory data $\mathbf{x}$, such that $\boldsymbol{\varepsilon}_l$ quantifies the disagreement between this prediction and the data. The same logic applies for $\boldsymbol{\varepsilon}_p$, which will be discussed further in the context of hierarchical models. Crucially, variational free energy is now written in terms of the sufficient statistics of $q_\lambda(\mathbf{z})$, i.e., the objective is now $\mathcal{F}(\mu, \mathbf{x})$ rather than $\mathcal{F}(\mathbf{z}, \mathbf{x})$ (see [30] for a full explanation).

The goal is now to find the value of $\mu$ which minimises $\mathcal{F}(\mu, \mathbf{x})$. This can be achieved through gradient descent (with some step size $\kappa$)

$$\dot{\mu} = -\kappa \ \frac{\partial \mathcal{F}}{\partial \mu} = -\kappa \ \left( \boldsymbol{\varepsilon}_p - \frac{\partial f_\theta(\mu)^\top}{\partial \mu} \boldsymbol{\varepsilon}_l \right) \tag{8}$$

While these updates may look complicated, they can be straightforwardly implemented in biologically-plausible networks composed of prediction units and error units (see [29]). The same scheme can be applied to learning the parameters of the generative model $\theta$, where the goal is now to find the value of $\theta$ which minimises $\mathcal{F}(\mu, \mathbf{x})$:

$$\dot{\theta} = -\kappa \ \frac{\partial \mathcal{F}}{\partial \theta} = -\kappa \ \left( \boldsymbol{\varepsilon}_l f_\theta(\mu)^\top \right) \tag{9}$$

where $\kappa$ is the learning rate. For each data point, $\mu$ is iteratively updated using Eq 8 until convergence, and then Eq 9 is updated based on the converged value of $\mu$, here denoted $\mu^*$ in considerations of HPC below. Crucially, Eq 9 can be implemented using simple Hebbian plasticity [29, 42]. Predictive coding is usually implemented in networks with $L$ hierarchical layers, where each layer tries to predict the activity of the layer below it (besides the lowest layer,

which predicts the data):

$$p(\mathbf{z}, \mathbf{x}) = p(\mathbf{x}|\mathbf{z}_1)p(\mathbf{z}_1|\mathbf{z}_2)\dots p(\mathbf{z}_{L-1}|\mathbf{z}_L)p(\mathbf{z}_L) \tag{10}$$

When considered as a neural process theory, predictive coding posits the existence of two neuronal populations: prediction units (which compute predictions) and prediction error units (which compute the difference between a predictions and the actual input) [80, 81]. To make predictions match input data, the dynamics described by Eqs 8 and 9 prescribe that prediction errors are minimised over time. This ensures that contextual information from superordinate layers are integrated into the representations of lower layers during inference, thereby helping to disambiguate ambiguous stimuli [9, 24, 82]. As variational free energy is equal to the sum of (precision-weighted) prediction errors (Eq 7), minimising prediction errors is equivalent to minimising variational free energy, and thus equivalent to performing inference on a generative model. It is for this reason that predictive coding is often considered a 'probabilistic' theory of learning (as the model variables can be read as sufficient statistics of posterior beliefs), even when all of the distributions are point mass or with fixed second-order statistics.

Predictive coding models the world in a top-down manner—e.g., it learns to predict features from objects, rather than predicting objects from features. It is this aspect which makes predictive coding *generative* (While predictive coding is usually considered to be unsupervised algorithm, it is straightforward to extend the scheme into a supervised setting [29, 83–86]. This can be achieved by turning the predictive coding network on its head, so that the model tries to generate hidden states (e.g., labels) from data.)—as it is able to generate data without any external input [33]. Inference is achieved by 'inverting' this model—i.e., going from data (stimuli) to hidden states (latent features). [81]. This inversion process is achieved by iteratively applying Eq 8 for a given number of steps. The generative nature of predictive coding means that it is able to take context into account during inference, and it can work with relatively small amounts of data. On the other hand, its inherently iterative nature is computationally costly and temporally slow, and—in addition—in standard implementations predictive coding is also *memoryless* [53], meaning that inference is repeated afresh for each stimulus, even if that stimulus has been encountered previously, Note that memoryless inference can be useful, in so far as it ignores any biases which may have been introduced by previous data points.

**Amortised inference.**   Amortised inference provides an alternative approach to performing variational inference which has recently gained prominence in machine learning [51]. Rather than optimising the variational parameters λ directly, amortised inference learns a function $f_\phi(\mathbf{x})$ which maps from the data to the variational parameters. The parameters $\phi$ of this function are then optimised over the whole dataset, rather than on a per-example basis. Once this function has been learned, inference is achieved via a single forward pass through $f_\phi(\mathbf{x})$, making amortised inference extremely efficient once learned. Amortised inference also retains some notion of memory [52], as the amortised parameters $\phi$ are shared across the available dataset. However, amortised inference cannot take contextual information into account, and suffers from the *amortisation* gap [54], which describes the decrease in performance incurred from sharing parameters across the dataset rather than optimizing them individually for each data point.

Formally, amortised inference solves the following optimisation problem:

$$\phi^* = \underset{\phi}{\arg\min}\, \mathbb{E}_{p(\mathcal{D})}\left[\mathbb{E}_{q_\lambda(\mathbf{z})}\left[\ln q_\lambda(\mathbf{z}) - \ln p_\theta(\mathbf{z}, \mathbf{x})\right]\right] \tag{11}$$

$$\text{where }\; \lambda = f_\phi(\mathbf{x})$$

where $\mathcal{D}$ is the available dataset, and $f_\phi(\mathbf{x})$ is the amortised function which maps from the data $\mathbf{x}$ to the variational parameters $\lambda$. The goal is then to optimise the parameters of this amortised function $\phi$ to minimize the variational free energy on average over the entire dataset.

In contrast with predictive coding, amortised inference is fundamentally a bottom-up process: it predicts objects from features. In the current context, $f_\phi(\mathbf{x})$ acts as a discriminative model which learns the parameters of the variational posterior $q(\mathbf{z})$. Moreover, during inference, the amortised parameters are fixed. Several methods have been proposed for implementing amortised inference [51, 87], and these usually rely on some form of backpropagation or stochastic sampling to compute or approximate average gradients of the amortisation function with respect to the free energy. In the following section, we present an extension of predictive coding which incorporates amortised inference in a biologically plausible manner.

## 2.3 Hybrid predictive coding

Our novel hybrid predictive coding (HPC) model combines both amortised and iterative inference into a single biologically plausible network architecture. Our claims of biological plausibility inherit from the predictive coding framework, which adheres to the principles of local computation and local plasticity [29–31, 64]. We do not claim this model to be any more (or less) biologically plausible than networks that meet these criteria. However, we do claim that our models have an increased degree of biological relevance due to the fact that they can account for empirical observations that evade explanation in a traditional predictive coding framework. We consider a model composed of $L$ hierarchical layers, where each layer $i$ is composed of a variable unit $\mu_i$ and an error unit $\boldsymbol{\varepsilon_i}$. In the same manner as predictive coding, each layer tries to predict the activity of the layer below it: $\mu_{i-1} = f_\theta(\mu_i)$, besides the lowest layer, which tries to predict the data $\mathbf{x}$ directly. The error units measure the disagreement between these predictions and the actual input $\boldsymbol{\varepsilon_i} = \mu_{i-1} - f_{\theta_i}(\mu_i)$.

In contrast with predictive coding, we assume an additional set of amortised parameters $\phi$, which correspond to bottom-up connections. The amortised parameters define non-linear functions which map activity at one layer to activity at the layer above, thereby implementing a bottom-up prediction: $\mu_i = f_{\phi_i}(\mu_{i-1})$. Here, the lowest layer operates directly on sensory data: $\mu_0 = f_{\phi_0}(\mathbf{x})$. We refer to these bottom-up functions $f_\phi(\cdot)$ as being *amortised* since they directly map from data $\mathbf{x}$ to the parameters of a distribution $\mu$. Crucially, both the top-down $f_\theta(\cdot)$ and bottom-up $f_\phi(\cdot)$ functions try to predict the same set of variables $\{\mu\}$.

Inference proceeds in two stages. The first 'amortised phase' takes the current sensory data $\mathbf{x}$ and propagates it up the hierarchy in a feedforward manner, utilising the amortised functions $f_\phi(\cdot)$. This produces a set of initial 'guesses' for each $\mu$ in the hierarchy and is analogous to the feedforward sweep observed in neuroscience [19] and in artificial neural network architectures [88]. The second 'iterative' phase refines each $\mu$ by generating top-down predictions and iteratively applying Eq 8.

Once inference is completed for the current input, generative and amortized parameters are updated via learning. In order to learn the generative parameters $\theta$, Eq 9 is applied to the converged values of $\mu^*$. In order to implement learning for the amortised parameters $\phi$, we introduce an additional set of error units $\boldsymbol{\varepsilon_i^\phi}$ which quantify the difference between the amortised predictions and the values of $\mu$ at convergence in the iterative phase, which we call $\mu^*$. These amortised prediction errors are defined as $\boldsymbol{\varepsilon_i^\phi} = \mu_i^* - f_\phi(\mu_{i-1})$. Given these errors, we can update the values for $\phi$ using Eq 9, where $f_\theta(\mu)$ is now replaced by $f_\phi(\mu)$. By constructing the model in this way, the process of amortised inference is symmetric to the original

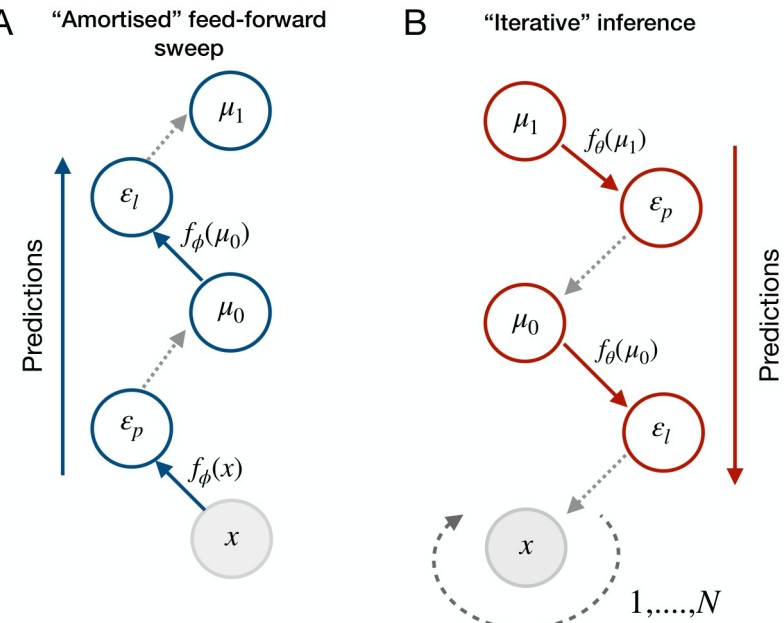

**Fig 2. Hybrid predictive coding combines two phases of inference as follows.** (**A**) At stimulus onset, data **x** is propagated up the hierarchy in a feedforward manner, utilising the amortised functions $f_{\phi(\cdot)}$. These predictions set the initial conditions for $\mu$, which parameterise posterior beliefs about the sensory data. These predictions are associated with error units that track the difference between variables at one level and the variables at the level above under transforms $f_\phi(\cdot)$. These errors are not utilised for inference but are used to update the amortised parameters $\phi$ during learning (weight updates). (**B**) The initial values for $\mu$ are then used to predict the activity at the layer below, transformed by the generative functions $f_\theta(\cdot)$. These predictions incur prediction errors $\varepsilon$, which are then used to update beliefs $\mu$. This process is repeated $N$ times, after which perceptual inference is complete.

predictive coding model, except that predictions now also flow in the opposite (bottom-up) direction.

A key aspect of the model is that the amortised predictions learn to predict beliefs at higher layers, *after the beliefs have been optimised by iterative inference*. In effect, the amortised predictions learn to 'shortcut' the costly process of iterative inference, allowing for fast and efficient mapping from data to refined beliefs. Fig 2 provides a schematic of the model, and Algorithm 1 presents the details of the inference and learning procedure.

**Algorithm 1** Hybrid predictive coding

```
Input: Generative parameters θ—Amortised parameters φ—Data x—Step
size κ—Learning rate α
Amortised Inference:
```
$\mu_0 = f_{\phi_0}(\mathbf{x})$
```
for i = 1...L - 1 do
```
   $\mu_{i+1} = f_{\phi_i}(\mu_i)$
```
end
Iterative Inference:
for optimisation iteration j = 1...N do
```
   $\boldsymbol{\varepsilon}_l = \mathbf{x} - f_{\theta_0}(\mu_0)$
   $\boldsymbol{\varepsilon}_p = \mu_0 - f_{\theta_1}(\mu_1)$
   $\dot{\mu}_0 = -\kappa \left( \boldsymbol{\varepsilon}_p - \frac{\partial f_\theta(\mu_0)^\top}{\partial \mu_0} \boldsymbol{\varepsilon}_l \right)$
```
   for i = 1...L do
```
      $\boldsymbol{\varepsilon}_l = \mu_{i-1} - f_{\theta_i}(\mu_i)$

$$\boldsymbol{\varepsilon}_p = \mu_i - f_{\theta_{i+1}}(\mu_{i+1})$$

$$\dot{\mu}_i = -\kappa \left( \boldsymbol{\varepsilon}_p - \frac{\partial f_\theta(\mu_i)^\top}{\partial \mu_i} \boldsymbol{\varepsilon}_l \right)$$

**end**

**end**

**Learning of Parameters:**

**for** $i$ = 0...$L$ **do**

$$\boldsymbol{\varepsilon}_l^\phi = \mu_{i+1}^* - f_{\phi_i}(\mu_i)$$

$$\dot{\theta}_i = -\alpha \left( \boldsymbol{\varepsilon}_l f_{\theta_i}(\mu_i)^\top \right)$$

$$\dot{\phi}_i = -\alpha \left( \boldsymbol{\varepsilon}_l^\phi f_{\phi_i}(\mu_i)^\top \right)$$

**end**

## 3 Results

To illustrate the performance of HPC, we present a series of simulations on the MNIST dataset, which consists of images of handwritten digits (from 0-9) and their corresponding labels. We first establish that HPC can simultaneously perform classification and generation tasks on the MNIST dataset. We then show that the model enables *fast inference*, in that the number of iterations required to achieve perceptual certainty reduces over repeated inference cycles. Moreover, we show that the novelty of the data adaptively modulates the number of iterations enabling more rapid adaptation to nonstationary environments and distribution shift. We then demonstrate the practical benefit of fast inference by plotting the accuracy of hybrid and standard predictive coding against the number of iterations, which demonstrates that our model can retain high performance with minimal iterations relative to standard predictive coding. To demonstrate the benefits of the top-down, generative component of our model, we compare the accuracy of HPC inference as a function of the dataset size and show that it can learn with substantially fewer data items than a purely amortized scheme. Finally, we investigate additional beneficial properties of our model. We show that the iterative inference phase can be accurately described as refining beliefs since it decreases the entropy of the initial amortized prediction. We show that our network can adaptively reduce computation time for well-learned stimuli but increase it again for novel data, as well as that combining the iterative and amortized components substantially reduces the number of inference iterations required throughout training.

### 3.1 Simulation details

The MNIST database consists of 50,000 training examples and 10,000 test examples. Each example is composed of an image and a corresponding label between 0 and 9, where each image is black and white and of size 28 x 28 pixels, which is fed into the predictive coding network via an input layer consisting of 784 nodes. In the context of both hybrid and standard predictive coding, labels are encoded as priors at the highest level of the hierarchy $L$. Specifically, the model's highest layer is composed of 10 nodes (one for each label). During training, these nodes are fixed to the corresponding label: the node which corresponds to the label is fixed at one, while the remaining nodes are set to zero. The bottom (sensory) layer is fixed to the current image during training. During testing, while the bottom layer remains fixed to the image, the highest layer is left unconstrained. To obtain a classification during testing, we return the label which corresponds to the top-layer node with the largest activity at the end of inference. For generation, we fix the top layer of the network to a desired label and leave the input nodes unconstrained. We then perform inference throughout the network until convergence and read out the inferred image at the bottom layer.

Both the hybrid and standard predictive coding models are composed of 4 layers of nodes ($L = 4$). The lowest layer, which is fixed during training and testing, comprises 784 nodes and corresponds to the current image. The next two layers are composed of 500 nodes each, and the highest layer is formed of 10 nodes, which correspond to the current label and are constrained during training. For both the hybrid and standard predictive coding models, the generative, top-down functions $f_\theta(\cdot)$ use `tanh` activation functions for all layer besides the lowest, which do not use an activation function. Weight normalisation, where the absolute values of the weights in a layer are divided by the mean of the weights of the layer, is used for the generative parameters $\theta$, which we found crucial for maintaining classification accuracy in the standard predictive coding network. For the amortised, bottom-up functions $f_\phi(\cdot)$ (only used in the hybrid model), a `tanh` activation is used for all layers besides the highest, which does not use an activation function. All weights are updated using the ADAM optimiser [89] with a learning rate of $\alpha = 0.01$, and $\kappa = 0.01$ is used for iterative inference. Th ADAM optimiser is used as a generic optimiser for updating parameters via gradient descent. While the ADAM optimiser is not biologically plausible, we found that the relative results between different models identified in the paper could be recreated with alternative optimisers, such that the specific optimiser used could be treated as a black box for our key results. Unless specified otherwise, we use $N = 100$ iterations during iterative inference. To demonstrate the adaptive computation properties of HPC we also use an adaptive threshold which cuts off inference if the average sum (across layers) of mean squared prediction errors is less than 0.005.

In contrast with standard presentations of MNIST results, we do not measure accuracy over entire epochs (e.g., the test set accuracy after the model has been trained on all 60,000 examples in a batched fashion) but instead measure accuracy as a function of batches. Specifically, we compute train and test accuracy after every 100 batches, where the batch size is set to 64 for all experiments. This strategy was chosen due to the speed at which our models converge (often within 600 batches, or approximately 38,000 examples), thereby allowing us to visualise convergence in a more fine-grained manner.

We calculate amortised accuracy by an amortised feedforward sweep and using the resulting activations of the final layer to calculate the loss, and the amortised model is then updated based on the bottom-up errors between results from this initial feedforward sweep and the final activation derived from iterative processing.

## 3.2 Unsupervised and supervised learning within a single algorithm

The first set of simulations illustrate that the model can perform both classification and generation simultaneously, meaning that it can naturally utilise both supervised and unsupervised learning signals. This property is desirable for a perceptual inference algorithm, since in many situations, training labels may only be available occasionally. The unsupervised capabilities of HPC derive from learning the top-down generative parameters $\theta$. In the absence of labels (or priors in the current framework), these parameters distil statistical regularities in the data, forming a generative model which can be used for various downstream tasks. The supervised learning capabilities derive from both the Bayesian model 'inversion' provided by iterative inference, and the bottom-up initialisation provided by amortised inference. Our model captures the relationship between data and labels in a probabilistic manner by constraining the highest layer's nodes to the relevant labels during training. It is important to note that the ability to utilise both supervised and unsupervised signals is not unique to hybrid predictive coding—standard predictive coding can also learn from both supervised and unsupervised signals. However, as we will show in the following experiments,

our hybrid architecture affords several additional benefits which are not provided by standard predictive coding.

We first demonstrate the classification accuracy and generative capabilities of HPC. We compare the results of hybrid and standard predictive coding on the MNIST dataset, and additionally, compare these results to the accuracy of the amortised component alone. Recall that the amortised accuracy corresponds to the initial 'best guess' provided by the amortised forward sweep, which is then refined by iterative inference to give the final accuracy of hybrid predictive coding. The present analysis therefore allow us to determine the influence that iterative inference has on the hybrid predictive coding model.

Results are shown in Fig 3. There is no significant difference between the classification accuracy of the hybrid and standard predictive coding (Fig 3A). This is to be expected, as the iterative inference procedure (shared by both hybrid and standard predictive coding) 'trains' the amortised component (as the amortized connections learn to minimize the prediction error between their own predicted beliefs and the beliefs eventually converged to in the process of iterative inference), meaning that the amortised component's accuracy cannot be higher than that provided by iterative inference alone. This 'training' can be seen in Fig 3A, where the amortised accuracy converges to the hybrid model's accuracy over time. The accuracy of amortised inference increases at a slower rate. This result is likely due to the fact that amortised inference is trying to predict the equilibrium point of iterative inference, which is

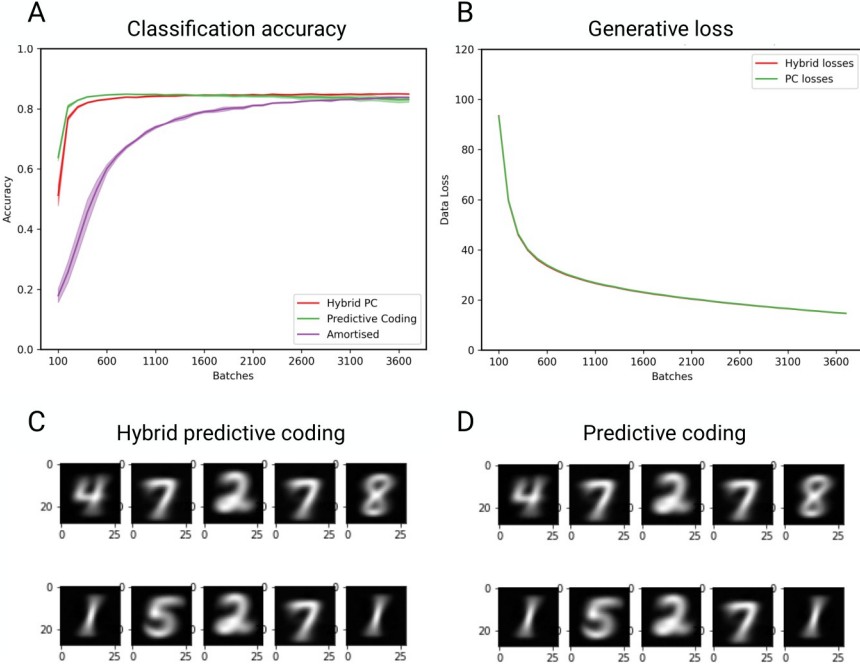

**Fig 3. Simultaneous classification and generation. (A)** Classification accuracy on the MNIST dataset for hybrid predictive coding, standard predictive coding and amortised inference. Each line is the average classification accuracy across three seeds; the shaded area corresponds to the standard deviation. The *x*-axis denotes the number of batches. **(B)** Generative loss. The panel shows the averaged mean-squared error between the lowest level of the hierarchy (which is fixed to the sensory data during testing) and the top-down predictions from the superordinate layer, plotted against batches, for HPC and standard PC. This metric provides a measure of how well each model is able to generate data. The seeds used are the same as those used in panel **(A)** (i.e., the data is from the same run). **(C)** Illustrative samples taken from HPC at the end of learning. These images are generated by activating a single nodes in the highest layer (corresponding to a single digit), and performing top-down predictions in a layer-wise fashion. The images correspond to the predicted nodes at the lowest layer. **(D)** As in **(C)** but for standard predictive coding.

changing as the parameters of the network converge. Similarly, Fig 3B shows that the hybrid and standard predictive coding models are equivalent in terms of their ability to generate data, and Fig 3C & 3D show that samples generated from each of these models are qualitatively similar. Again, these results are expected, as the process of amortised inference should have little influence on the learning of the generative parameters. As illustrated later, the predictive coding and hybrid predictive coding architectures possess a powerful capacity to learn from limited data samples, which may account for the superior performance of the predictive coding network during initial training stages. Nevertheless, the predictive coding network's accuracy shows a marginal decline due to training procedure instabilities that are absent in hybrid predictive coding. It is worth noting that the asymptotic performance is somewhat lower than usually reported on the MNIST dataset. This discrepancy is explained by the fact that we are using models that are fundamentally generative, i.e., their objective is to generate the data, not perform classification.

## 3.3 Fast inference

In the previous section, we demonstrated that hybrid predictive coding retains standard predictive coding's classification accuracy and generative capabilities. We next show that the inclusion of a bottom-up, amortised component facilitates *fast inference*, by which we mean the ability to reach some level of perceptual certainty in a reduced number of iterations. To operationalise perceptual certainty, we introduce an arbitrary threshold (here 0.005) and stop iterative inference once the sum of average squared prediction errors (i.e., the free energy) has fallen below this threshold. The averaged prediction error can be thought of as a proxy for perceptual certainty because it is equivalent to variational free energy in the current context, thereby providing a principled measure of model fit since, after the minimization of the variational free energy is complete, it will come to approximate the log model evidence for a particular setting of the generative model parameters. Continuing with the same experimental setup, Fig 4B shows that the number of iterations required to reach perceptual certainty decreases over batches. Once converged, asymptotic accuracy is achieved without requiring any iterations at all, meaning that accurate perceptual beliefs are furnished through a single amortised forward sweep without the need for any expensive iterative inference steps, thus furnishing rapid and computationally cheap perception for commonly encountered data. Fig 4A demonstrates that this reduction in iterations has no detrimental effect on the accuracy of hybrid predictive coding.

To demonstrate the practical benefit of fast inference, we compare the accuracy of hybrid and standard predictive coding when using a fixed number of iterations (i.e., no 'perceptual certainty' threshold). Specifically, we compare classification accuracies when using 10, 25, 50 and 100 iterations for iterative inference. Results are shown in Fig 5, using the same simulation setup as in previous experiments, apart from the number of iterations. Hybrid predictive coding can obtain equivalent performance with a little as 10 iterations (Fig 5A), whereas standard predictive coding fails to learn at all under these conditions. At 25 iterations (Fig 5B), we see that the performance of standard predictive coding slowly decreases over batches. Notably, we observed no such general performance decrease for hybrid predictive coding suggesting that the amortized bottom up connection help to 'stabilize' learning in the hybrid network. Together, these results further illustrate that hybrid predictive coding facilitates fast inference by bypassing the need for costly iterative inference when amortized inferences are sufficiently accurate.

Another interesting phenomenon is the relative difference in accuracy between the full hybrid model and its amortised component as a function of iterations. When the number of iterations is lower (e.g., Fig 5A), the relative difference between these accuracies is far less

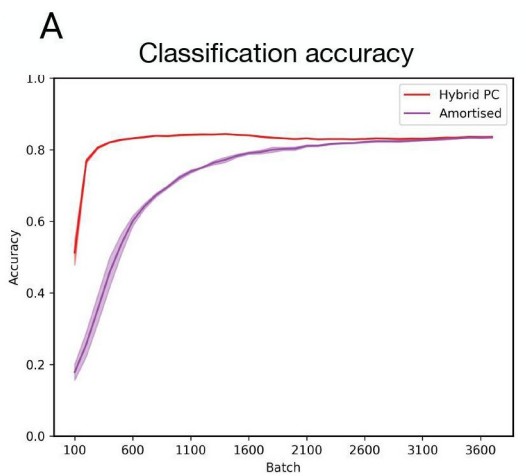
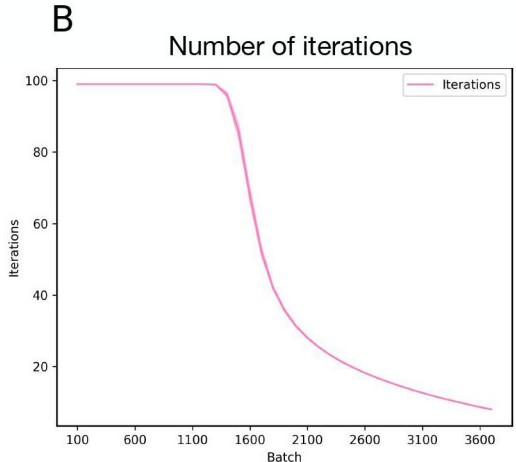

**Fig 4. Fast inference.** (A) Classification accuracy of the hybrid predictive coding model and the bottom-up, amortised predictions as a function of number of batches. The asymptotic convergence demonstrates that placing an uncertainty-aware threshold on the number of iterations has no influence on (asymptotic) model performance. Plotted are average accuracies over 5 seeds and shaded regions are the standard deviation. (B) Average number of iterations (for iterative inference) as a function of test batch. Amortised predictions provide increasingly accurate estimates of model variables, reducing the need for costly iterative inference.

pronounced since the accuracy of the pure iterative inference predictive coding network is unstable and decreases over time when there are an insufficient number of inference iterations. The amortized feedforward pass in the hybrid model, by providing an approximately 'correct' initialization, enables the network to furnish accurate beliefs within many fewer inference steps. These results suggest that, when the number of iterations is limited, amortised learning progresses at a faster rate and, in the limit, can enable progress even in situations where purely iterative learning fails.

We identify several unintuitive results that require further explanation, and which may be suitable predictions for empirical explorations. First, the accuracy of the amortised network improves slower as the number of inference iterations increases. We hypothesise that this is due to the targets for the amortised network being more difficult to predict (due to an increased number of iterations, leading to increased accuracy). Second, the amortised component can achieve high accuracy while the iterative component achieves low accuracy. This is due to the final layer loss—which is a function of the ground truth labels and which provides a learning signal in lieu of accurate iterative inference. Finally, we observe instabilities in the predictive coding network that occur at a low number of iterations, leading to a decline in accuracy. We hypothesize that these result from instabilities in the training procedure, which crucially are not found in the full hybrid network [90].

### 3.4 Learning with limited data

Having illustrated the benefits of incorporating a bottom-up, amortised component into predictive coding, we next consider the benefits of the top-down, iterative component of the model. With slight modifications [86], the amortised predictions of our model can approximate the backpropagation algorithm, meaning that the bottom-up connectivity implements something akin to a multi-layer perceptron. This might lead to the worry that the top-down component of hybrid predictive coding is superfluous, and a purely bottom-up scheme would suffice. There are several reasons why this is not the case. First, the top-down, generative

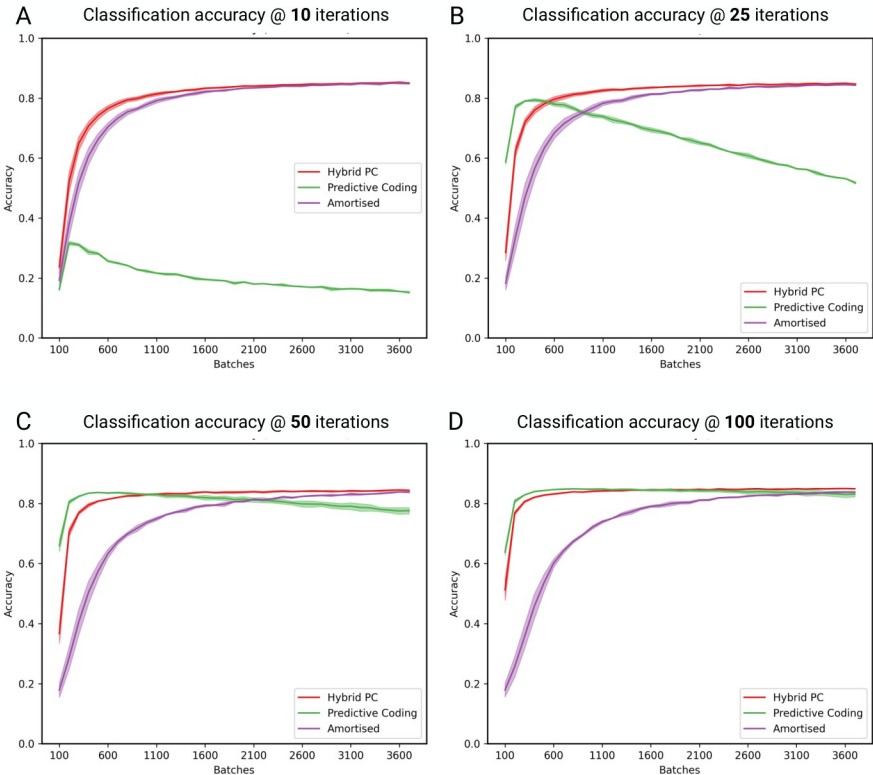

**Fig 5. Classification accuracy under fixed iterations. (A)** 10 iterations. The accuracy of HPC and the amortised predictions is mostly unaffected by the reduced number of iterations, whereas standard predictive coding fails to classify at all. **(B)** 25 iterations. The classification accuracy of standard predictive coding slowly decreases over batches, illustrating a common pathology observed in these simulations. **(C)** 50 iterations. Standard predictive coding approximately matches the performance of hybrid predictive coding, but begins to decline later in training. **(D)** 100 iterations. There are no significant differences between the accuracies of hybrid and standard predictive coding. Together, these results demonstrate that hybrid predictive coding enables effective inference and maintains higher performance with a substantially fewer amount of inference iterations required than standard predictive coding. Plotted are mean accuracies over 5 random network initializations. Shaded areas are the standard deviation.

component provides the training data from which the amortised component learns. Second, learning a generative model is generally more data-efficient compared to learning discriminative models. Here, we demonstrate the speed at which the network learns by plotting accuracy as a function of the amount of data the network has been exposed to. Specifically, we compare the accuracy of hybrid predictive coding compared to the amortised predictions using datasets with 100, 500, 1000 and 5000 examples (recall that the full dataset contains 60,000 examples). Note that we still use the complete test set of 10,000 images for testing. As Fig 6 shows, hybrid predictive coding retains good performance with as few as 100 training examples. By contrast, the speed at which the amortised predictions converge is significantly affected by the dataset size, such that the amortised predictions give poor accuracy in low data regimes. These results show that incorporating a top-down, generative component substantially increases the speed at which the network learns. It is also notable that performance of HPC is not negatively affected by the poor accuracy of the amortised predictions, again demonstrating an adaptive trade-off between amortised and iterative inference which allows for the iterative inference procedure to overcome a poor initialization by the amortized predictions.

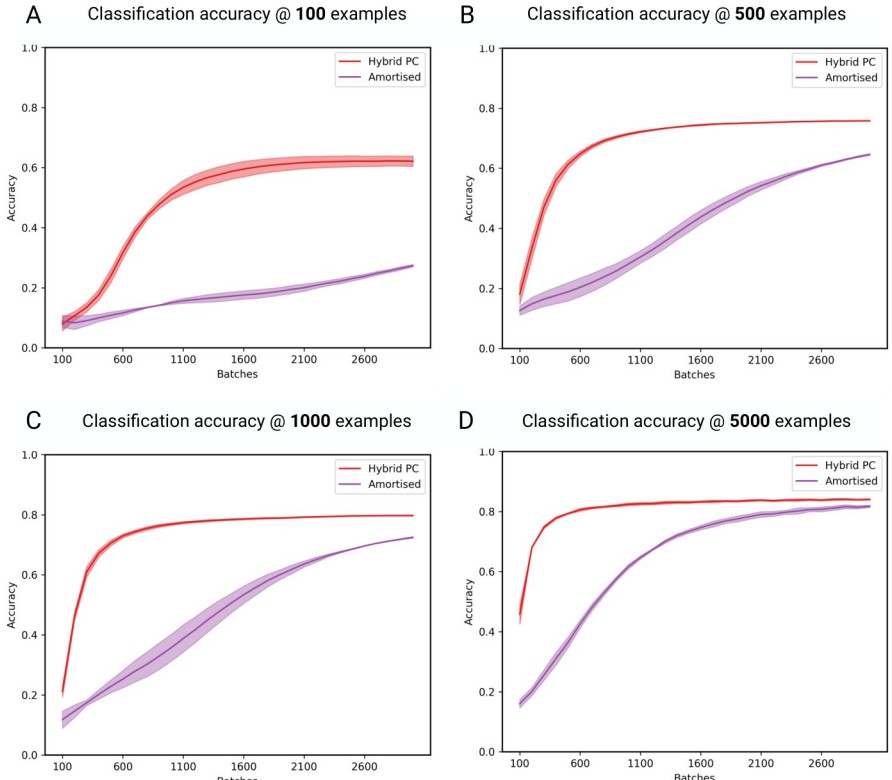

**Fig 6. Accuracy as a function of dataset size. (A)** 100 examples. The accuracy of hybrid predictive coding is lower than with the full dataset, but still high given the relatively small amount of data the network has been exposed to (0.17 percent). The accuracy of the amortised predictions is significantly worse **(B)** 500 examples **(C)** 1000 examples. **(D)** 5000 examples. Together, these results demonstrate that bottom-up, amortised inference is far more sensitive to the time spent training, compared to the full hybrid architecture. Importantly, the poor performance of amortised inference at the start of learning does not negatively impact the speed at which iterative inference learns. Plotted are the mean accuracies over 5 seeds. Shaded areas represent the standard deviation.

## 3.5 Additional properties of HPC

To gain further intuition for the functioning of the HPC model, we investigate several other properties of the model. Firstly, we investigate the degree to which the model's own uncertainty evolves during the inference process. We quantify the model's uncertainty as to the correct label by the entropy of its distribution over the predicted labels $0 - 9$. This is achieved by normalizing the values for the final layer to obtain an output probability distribution and calculating the Shannon entropy of this distribution. In Fig 7A, we show that this entropy begins high and monotonically decreases through an inference iteration, thus suggesting that in general the iterative inference process serves to sharpen and clarify beliefs. Secondly, we investigated in more detail the computational savings the hybrid model achieves through its accurate initialization of the iterative inference via the amortized model. In Fig 7B, we plotted the number of inference iterations utilized for each batch during learning for the hybrid and the standard predictive coding model. We used an early stopping criterion by which iterations are terminated when an absolute error threshold $\beta$ is reached. We see that for HPC the number of iterations required rapidly drops off during learning, due to the successful bootstrapping of the amortized model while for standard predictive coding the number of inference iterations only start declining towards the end of training when the network weights have adapted to become good at explaining the data. Since the iterative inference steps are the main

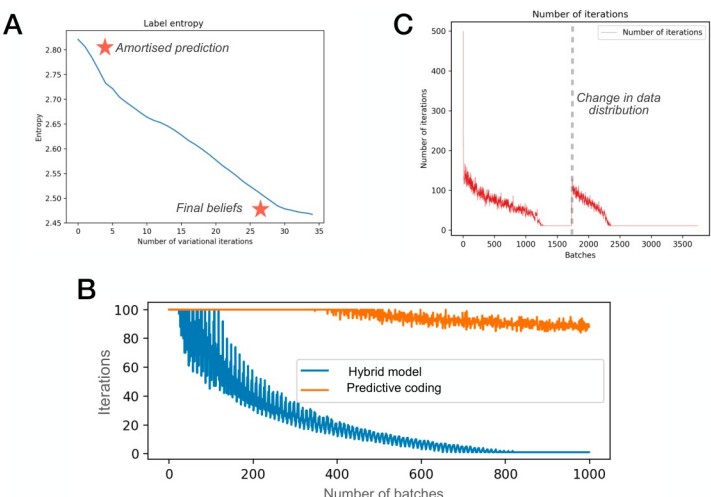

**Fig 7. Additional properties of the HPC model.** (**A**) Example evolution of the label entropy over the course of an inference phase. The initial amortized guess has relatively high entropy (uncertainty over labels) which progressively reduces during iterative inference. This is consistent with the viewpoint that the iterative inference phase refines the initial amortized guesses. (**B**) The number of inference steps required over an example training run. Due to the superior initialization provided by the amortized connections, far fewer iterative inference steps are required. (**C**) Adaptive computation time based on task difficulty. On a well learned task, the number of inference iterations required decays towards 0. However, when there is a change in data distribution, additional iterative inference iterations are adaptively utilized to classify the new, more challenging, stimuli.

computational cost of the model (the weight updates cost at most the same as a single inference iteration) HPC achieves a substantial computational saving over standard predictive coding while also obtaining equal or higher performance, as shown in previous figures.

Finally, the ability for amortised inference to 'shortcut' iterative inference is facilitated by the stationary data distribution used so far. Therefore, we investigated whether changes in the data distribution modulate the number of iterations to reach perceptual certainty. To do this, we split the dataset into two halves—one composed of labels 0 through 5, and the other composed of labels 5 through 9. Initially, we train and test using only the first of the two datasets. As shown in Fig 7C, the number of iterations (for iterative inference) decreases towards zero during this period. To enact a change in data distribution for the second half of training and testing, we utilise only the second half of the dataset. As Fig 7C shows, this dramatically increases the number of iterations required to reach perceptual certainty. This is because the bottom-up, amortised predictions have not learned the predict the relevant model variables, leading to a poor initialisation and an increase in prediction error. Crucially, this increase in iterations is automatically modulated by the data's novelty (or formally, the log-likelihood of the data under the generative model), highlighting that HPC provides a principled mechanism for trading off speed and accuracy during perceptual processing. Crucially, the relationship between prediction errors and the final accuracy of the beliefs is non-trivial; it is possible for a randomly initialised PC network to achieve zero prediction error in a short number of iterations, and a fully converged PC network to incur a large amount of prediction error after several iterations, while still achieving accurate beliefs. This is demonstrated in experiment Fig 7C, where directly after the change in data distribution, the HPC model cannot classify the previously unseen digits but still manages to achieve perceptual certainty after several iterations. In this context, perceptual certainty refers to the equilibrium point of inference, which may or may not reflect accurate beliefs. As such, this plot tracks the number of iterations required to go from the initial state to the equilibrium state of the network. As demonstrated in Fig 7B,

this is mostly facilitated by the role of the amortised network, which provides an initial state closer to the final equilibrium. However, it is partly facilitated by learning the parameters of the generative network (orange line), as learning changes the optimisation landscape that the iterative process operates on, enabling activity to converge in fewer iterations.

## 4 Discussion

The notion that the brain performs or approximates Bayesian inference has gained significant traction in recent years [7, 8, 68, 69, 91–93]. At the same time, the predictive coding architecture has gained prominence as a process theory which could provide a neurobiological implementation of approximate Bayesian inference [8, 31, 37, 42, 84]. However, there are many ways in which Bayesian inference and learning could be implemented or approximated in the brain. In this paper, we have described a novel architecture—hybrid predictive coding (HPC)—which combines amortised and iterative inference in a principled manner to achieve perceptual inference. In this biologically plausible architecture, predictions (and prediction errors) flow in both top-down and bottom-up directions. The top-down generative aspect of the model allows effective inference in low data regimes through relatively slow iterative inference. The bottom-up amortised (discriminative) aspect allows fast inference in stable data regimes. Hybrid predictive coding inherently balances the contributions of these two components in a data-driven 'uncertainty aware' fashion, so that the model inherits the benefits of both. As well as offering a novel machine learning architecture, hybrid predictive coding provides a powerful computational lens through which to understand different forms of visual perception—in particular, differences between fast context-free perception, such as gist perception, and slow, context-sensitive perception, such as detailed object recognition. Further work under the banner of 'computational neurophenomenology' (e.g., [94, 95]) is needed to examine this proposition in more detail, and in particular to explore how aspects of iterative/amortized inference correspond respectively to aspects of detailed/gist perceptual phenomenology.

To illustrate HPC, we have presented a number of simulations demonstrating that incorporating bottom-up, amortised predictions into a combined HPC architecture retains the benefits afforded by standard predictive coding (Fig 3), such as the ability to learn in both a supervised and unsupervised manner and work efficiently in low data regimes (Fig 6), while additionally enabling fast inference, a method for shortcutting the costly and time-consuming process of iteratively minimising prediction errors (Fig 4). Crucially, we have shown that the trade-off between fast, bottom-up, amortised inference and slow, top-down iterative inference is automatically modulated based on the model's uncertainty about the data, enabling the model to utilise the benefits of both in an adaptive manner (Fig 7).

### 4.1 Properties of hybrid predictive coding

Previous work has described how amortised inference can be construed as *learning to infer*, as it seeks to learn a function which performs inference, rather than performing inference itself [96] directly. Building upon this insight, hybrid predictive coding can be considered an example of learning to infer *through inference*, as the learning signal for amortised inference is provided by the outcome of iterative inference. This mechanism provides a straightforward way to generate 'correct' layerwise targets internally for the amortized inference process to treat as supervised targets which enables the inference process to be bootstrapped rapidly while only using local learning rules in an unsupervised fashion.

Moreover, by providing a rapid initial 'best guess' at representations [9, 16, 97], amortised inference provides informed initial conditions for the subsequent phase of iterative inference, which decreases the number of iterations required to obtain accurate (or useful) beliefs.

This dynamic enables rapid responses to novel events while also allowing for improved context-sensitive disambiguation of uncertain stimuli due to the refining effect of additional iterative inference steps. This means that, over time, in statistically stable (stationary) environments, bottom up predictions will learn to accurately capture perceptual beliefs, eschewing the need for costly iterative inference and therefore achieving substantial computational savings. However, if the environment changes, amortization will lead to errors which can be corrected by the iterative inference, allowing a rapid and context-sensitive response at the cost of greater computational expense during the period of adaptation to the novel environment.

Our model also naturally balances top-down and bottom-up contributions to perceptual inference. The degree to which bottom-up and top-down predictions contribute to the final beliefs is adaptively modulated based on their relative uncertainty, equipping the model with a basic notion of 'confidence', 'uncertainty monitoring', or 'calibration' [98]. By combining amortised and iterative inference within a single model, the resulting scheme naturally facilitates an adaptive trade-off between speed and accuracy, which is not possible when utilising either method of inference in isolation.

By building upon predictive coding, our model retains the generative capabilities of this algorithm [64, 99], facilitating broader generalisation and unsupervised learning. But the inclusion of bottom-up predictions also introduces a discriminative element to our model, as these bottom-up predictions map directly from data to beliefs. Because the model retains the neural architecture and Hebbian updates of predictive coding, it inherits the wealth of neurophysiological evidence that has accrued in favour of this theory [41]. Crucially, both the amortised and iterative inference schemes use the same learning rule, allowing us to posit a unified learning algorithm that underwrites both forms of inference. Moreover, by introducing a bottom-up process of amortised inference, our model provides a compelling account of feed-forward activity during perception and accords neatly with empirical results that demonstrate that core object recognition and "gist" perception occur on time scales which preclude the use of recurrent dynamics [48–50].

More generally, our model fits with the growing body of work which aims to describe and understand the relative roles of feed-forward and recurrent activity in the brain. Specifically, abundant evidence suggests that recurrent activity aids the processing of ambiguous, occluded and degraded objects, such that the degree of recurrent activity correlates with the ambiguity of the stimulus [100–105]. Our model can straightforwardly account for these results, as it proposes that recurrent activity implements a process of iterative inference, which refines the beliefs initialised by amortised inference until an accuracy threshold is reached, which would take more iterations for more ambiguous stimuli thus leading to longer response times and more effortful processing.

The model can also explain the effectiveness of the initial feed-forward sweep in core object recognition [19, 106], as simple types of object recognition can be accounted sufficiently for by amortised inference [107–110]. Our work builds upon recent machine learning architectures which have combined amortised and iterative inference in different ways, such as the semi-amortised variational autoencoders [111] and iterative amortised inference [46]. In those previous works, variational autoencoders [51] are augmented with a separate iterative inference phase, helping overcome the shortcomings of amortised inference [54]. Our model is distinguished from them by additionally developing a biologically plausible architecture closely matching neurophysiological observations and framed within the powerful context of predictive coding. Similarly, our model builds upon architectures that incorporate generative feedback into feed-forward neural networks [112]. These previous results demonstrate that generative feedback enables robustness to noise and adversarial attacks. However, these

models are trained through backpropagation and are not formed in a biologically plausible manner. Finally, our work builds upon efforts to combine the relative benefits of generative and discriminative models [113].

## 4.2 The feedforward sweep and beyond

In visual neuroscience, object recognition is often separated into two distinct phases [18, 114]: an initial 'feedforward' sweep (lasting around 150ms) [14, 19, 114], in which sensory data is rapidly propagated up the visual hierarchy in a feedforward manner, and a subsequent stage of recurrent processing which persists over longer periods [16]. It has been argued that feedforward processing provides coarse-grained representations sufficient for core object recognition and so-called 'gist' perception, while recurrent processing finesses these representations by integrating contextual information [9, 16, 17, 115, 116] and allowing for the resolution of initial ambiguity or uncertainty.

This account of perception is remarkably consistent with our proposed model. In this context, the feedforward sweep corresponds to the amortised 'best guess' at perceptual beliefs, which is implemented by feedforward connectivity in our model. Crucially, these amortised predictions are insensitive to current context, as they map directly from data to beliefs. Moreover, amortised predictions suffer from the amortisation gap [54], which arises when parameters are shared across the whole dataset rather than optimized individually for each data point. Taken together, these considerations suggest that the beliefs furnished by amortised inference could lack the ability to successfully underlie perception for challenging (e.g., weak sensory signals), ambiguous, or otherwise unusual situations which require prior contextual knowledge to successfully parse—a result consistent with empirical evidence about the respective roles of feedforward and feedback processing in perception [18, 19, 117]. In line with this view, in our model we see a slower increase in accuracy for amortised inference, compared to the full hybrid predictive coding architecture, for small datasets. The implication here is that these small datasets cannot be modelled well with purely amortized inference, but can be modelled well by the combination of both iterative and amortized inference components. In addition, our model casts the recurrent processing in the visual system as a process of iterative inference, where beliefs are iteratively refined based on top-down predictions interacting with bottom-up beliefs and with sensory input. This iterative refinement integrates contextual information across multiple layers and slowly reduces the ambiguity in perceptual beliefs as the inference process converges to the best explanation. Again, this perspective is in accordance with neuro-biologically-informed views suggesting that recurrent processing refines the representations generated during the feedforward sweep [17, 47, 105].

Our model makes several predictions which have been corroborated by empirical evidence. For instance, our model predicts that the amount of recurrent processing will correlate with the difficulty of perceptual processing tasks. In line with this, [116] reported human neuroimaging data suggesting increased recurrent processing for more challenging perceptual tasks. In addition, our model predicts that perceptual difficulty should modulate the relative influence of bottom-up and top-down processing, as has been observed in experimental data [118].

While there have been several proposals for how feedforward and recurrent activity may be integrated in the brain, our model is the first to combine these into a common and biologically plausible predictive-coding architecture. Doing so provides a principled arbitration between speed and accuracy in perceptual processing [119]. In our model, recurrent dynamics are driven by prediction errors. When prediction errors are minimised (i.e., predictions are accurate), recurrent activity is suppressed. This means that when amortised predictions generates accurate beliefs, there will be no prediction errors and no recurrent activity. Alternatively,

when amortised predictions generate inaccurate beliefs, prediction errors will be large and iterations of recurrent activity are engaged to finesse beliefs. Crucially, this arbitration arises naturally from the representations within the model. In summary, our model provides a plausible account for both feedforward and recurrent activity in the brain, which can be related to distinct forms of visual perception.

Empirical studies that explore the interaction between feedforward and recurrent activity typically utilize a stimulus-response paradigm, in which a single, static stimulus is introduced, and brain responses are recorded. In such paradigms, distinct feedforward and recurrent activity signatures can be observed [120]. The present study follows this approach by conducting inference on a single, static image and simulating a feedforward pass, followed by recurrent iterations. However, the brain is exposed to continuous dynamic stimuli in natural environments, leading to a less distinct separation between feedforward and recurrent activity. In future research, we aim to expand this model to dynamic stimuli and investigate whether the hybrid predictive coding scheme can be replicated without artificially separating the two stages, such as through interleaved recurrent updates at varying temporal resolutions.

One potential criticism of the hybrid predictive coding model is that it predicts recurrent feedback activity going to zero in static environments. However, the human brain never encounters fully static data—thanks to environmental fluctuations, internal movements such as saccades—or even just ongoing internal neural fluctuations (and there is some circumstantial evidence from phenomena like retinal stabilization [121, 122] that brain activity and perception *do* go to zero for fixed stimuli). This means that amortised, feedforward predictions should never fully explain the data, so there is always a continual flow of prediction error signals that recurrent activity must explain away.

Interestingly, some prominent theories of consciousness emphasize recurrent (i.e., top-down) signalling as critical for conscious perception ([123, 124], for a review see [125]). While these theories are not always interpreted in terms of perceptual inference, it is natural to do so [126]. This however raises an interesting possibility: could it be that conscious perception depends not on top-down signalling defined anatomically (as in standard recurrent theories) but, instead, defined functionally, in terms of their role in conveying predictions (as opposed to prediction errors)? This would predict that the relationship between conscious status and (anatomical) top-down signalling would depend on the details of the perceptual task and/or aspect of phenomenology being assayed. Future work could usefully investigate this possibility.

### 4.3 Predictive coding

Predictive coding has been shown to explain a diverse range of perceptual phenomena, such as end stopping [6], bistable perception [82, 127], repetition suppression [128] and illusory motion [129] (see [41] for more). Moreover, recent work has demonstrated that predictive coding provides a local approximation to backpropagation; the algorithm underwriting many of the recent successes in machine learning [84, 86, 130]. As such, it presents one of the leading theories for perception and learning in the brain [131] and by building on the predictive coding framework, our model inherits the wealth of empirical evidence that has been gathered in its favour [41, 42, 132, 133].

While predictive coding has emerged as a promising candidate for understanding cortical function, its iterative nature fits poorly with some established facts about visual perception. Prominent among these is that the visual system can reliably extract a range of features within 150ms of stimulus onset [14, 48–50], a timescale which would seem to preclude the presence of multiple iterations of recurrent dynamics, and in turn, the use of iterative inference. In

short, predictive coding struggles to account for rapid "gist" perception [134, 135], an essential component of visual perception. To overcome this shortcoming, our model augments predictive coding with additional bottom-up connectivity, which provides amortised estimates of perceptual beliefs using a single forward pass. The feedforward nature of the amortised connections means that representations can be extracted rapidly without relying on recurrent activity [106]. Although predictions are generally associated with top-down recurrent processing, this bottom-up forward pass can also be interpreted as its own kind of prediction [23]—predicting beliefs directly from data—with its own set of prediction errors that are minimized during learning. This perspective lets us see our model as simply performing bidirectional prediction and prediction error minimization on a unified objective. It is intriguing to consider, from the perspective of "computational phenomenology", whether the distinct phenomenological character of gist perception (in which an overall context is experienced), compared to detailed focal perception (in which fine details of, for example, visual objects) can be understood in terms of these differing forms of perceptual prediction.

## 4.4 Generative and discriminative models

In machine learning, a common distinction is made between generative and discriminative methods [136]. Generative methods learn a joint distribution over sensory data and hidden causes $p(\mathbf{x}, \mathbf{z})$, whereas discriminative methods learn a conditional mapping from data to hidden causes $p(\mathbf{z}|\mathbf{x})$. It is well established that generative methods are more efficient in low data regimes [137], can be used for a wider range of downstream tasks [138], and enable better generalisation [139]. On the other hand, discriminative methods are more efficient when the goal is to predict hidden states, and generally reach higher asymptotic performance [140]. This is because discriminative methods only learn about features relevant for discrimination, whereas generative methods learn about the data distribution itself. In general, generative methods enable unsupervised learning, where hidden states are not known in advance, whereas discriminative methods utilize supervised learning with a known set of ultimate hidden states—the labels.

Our model combines generative and discriminative components within a single architecture. The top-down connectivity implements a generative model, whereas the bottom-up connectivity implements a discriminative classifier (where the labels are now the hidden states of the generative model). The model thus retains the benefits of generative approaches, while also incorporating the benefits of discriminative learning. In contrast to previous proposals which combine generative and discriminative learning [112, 113, 141–143], our model operates within the biologically plausible scheme of predictive coding and automatically arbitrates the relative influence based on the uncertainty of bottom-up and top-down predictions.

An additional benefit of combining generative and discriminative methods is that it enables *generative replay* [144–146]. This describes the process of generating fake data (using some generative model) which is then used for downstream tasks. For instance, the generated data can be used to overcome 'catastrophic forgetting' [147] and enable continual learning [146]. In the context of our work, the generative model can be used to produce data from which the amortised component can learn. This has the benefit of reducing the amount of real-world data required for accurate inference. Another exciting possibility, arising directly from the HPC architecture, is using the discriminative model to generate hidden states which can then be used to train the generative model. These opportunities are afforded by the bi-directional modelling at the heart of our architecture and have been explored extensively in the reinforcement learning literature where the idea of using a learnt model to train an amortized policy or vice versa is common [45, 59, 148]. Finally, the fact that the generative and discriminative

connections are implemented in a layer-wise fashion means that replay can operate on a layer-by-layer basis. In brief, the generative model can help train the discriminative model which can help train the generative model. One hypothesis is that this process happens during sleep, when the brain is detached from veridical data [149]. An open question remains as to whether hybrid predictive coding learns *different* representations, relative to standard predictive coding, or whether the observed effects just influence the manner in which the two networks learn. We hope to explore this question in future work.

## 4.5 Conclusion

We have proposed a novel predictive coding architecture—*hybrid predictive coding*—which combines iterative and amortized inference techniques to jointly optimize a single unified energy function using only local Hebbian updates. We demonstrate that this architecture enables both rapid and computationally cheap inference when in stable environments where a good model can be learned, while also flexibly providing more accurate inference in sparse data or early learning regimes. The results of our model additionally suggest that hybrid predictive coding will behave favorably when faced with more difficult stimuli, due to it being able to flexibly deploy additional computation as a function of experience with a given stimulus. Our hybrid model also can learn rapidly from small datasets, and is inherently able to detect its own uncertainty and adaptively respond to changing environments. Hybrid predictive coding thus provides a novel model that unifies disparate perspectives on the biological role of feedforward sweeps and recurrent activity in the visual cortex, offering a parsimonious explanation for several different and often opposing perspectives, while reproducing several well-known experimental effects in this field and potentially even accounting for distinct aspects of visual phenomenology.

## Acknowledgments

AT was funded by a PhD studentship from the Dr. Mortimer and Theresa Sackler Foundation and the School of Engineering and Informatics at the University of Sussex. BM was supported by an EPSRC PhD studentship. AKS is supported by ERC Advanced Investigator Grant CONSCIOUS, grant number 101012954, and by the Dr Mortimer and Theresa Sackler Foundation. CLB is supported by BBRSC grant number BB/P022197/1 and by Joint Research with the National Institutes of Natural Sciences (NINS), Japan, program No. 01112005. AKS is additionally grateful to the Canadian Institute for Advanced Research (CIFAR, Program on Brain, Mind, and Consciousness). AT would also like to recognize the support of the Verses Research Lab. BM was additionally supported by Rafal Bogacz with BBSRC grant BB/s006338/1 and MRC grant MC_UU_00003/1.

## Author Contributions

**Conceptualization:** Alexander Tscshantz, Beren Millidge, Anil K. Seth, Christopher L. Buckley.

**Data curation:** Alexander Tscshantz.

**Formal analysis:** Alexander Tscshantz, Beren Millidge.

**Funding acquisition:** Anil K. Seth, Christopher L. Buckley.

**Investigation:** Alexander Tscshantz, Beren Millidge.

**Project administration:** Christopher L. Buckley.

**Software:** Alexander Tscshantz, Beren Millidge.

**Supervision:** Christopher L. Buckley.

**Visualization:** Alexander Tscshantz, Beren Millidge.

**Writing – original draft:** Alexander Tscshantz, Beren Millidge.

**Writing – review & editing:** Alexander Tscshantz, Beren Millidge, Anil K. Seth, Christopher L. Buckley.

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
