## [Decision Letter · Decision Letter 0]

5 Dec 2022

Dear Dr. Millidge,

Thank you very much for submitting your manuscript "Hybrid Predictive Coding: Inferring, Fast and Slow" for consideration at PLOS Computational Biology.

As with all papers reviewed by the journal, your manuscript was reviewed by members of the editorial board and by several independent reviewers. In light of the reviews (below this email), we would like to invite the resubmission of a significantly-revised version that takes into account the reviewers' comments.

We cannot make any decision about publication until we have seen the revised manuscript and your response to the reviewers' comments. Your revised manuscript is also likely to be sent to reviewers for further evaluation.

Sincerely,

Ulrik R. Beierholm

Academic Editor

PLOS Computational Biology

Marieke van Vugt

Section Editor

PLOS Computational Biology

Reviewer's Responses to Questions

**Comments to the Authors:**

Reviewer #1: This paper outlines an extension to the predictive coding (PC) architecture, adding bottom-up prediction connections. This new model, which now has prediction projections running in both directions, overcomes the slow and relatively ineffective inference performance of previous PC models, and offers a better fit to the empirically observed speed of “gist” perception.

The paper is well written, and describes important work that will be of broad interest in the fields of cognitive neuroscience, and artificial intelligence.

One of my main concerns is about its novelty. A similar model, that also included prediction connections going in both directions, was proposed in

D. Xu, A. Clappison, C. Seth and J. Orchard, "Symmetric Predictive Estimator for Biologically Plausible Neural Learning," in IEEE Transactions on Neural Networks and Learning Systems, vol. 29, no. 9, pp. 4140-4151, Sept. 2018.

The authors should disambiguate how their model is similar and/or different.

My other main issue concerns the description of their model. A number of important design aspects were unclear, or not easy to decipher. For example, on line 252, the authors state that they “introduce an additional set of error units”. However, figure 2, which illustrates their network architecture, does not show these error nodes (although it DOES show the error nodes of the original PC network).

Moreover, line 471 says that “the amortised and iterative inference schemes use identical connectivity and weight updates”. I don’t understand how the amortised and iterative inference schemes use the same weight updates. The iterative inference process uses f_theta, while the amortised inference uses f_phi. Algorithm 1 shows different weight updates for those two functions. Am I missing something?

The abstract claims that “the resulting scheme can be implemented in a biologically plausible neural architecture”. Without clarification of the issues above, it is difficult for me to assess that claim.

On line 247, it says “Inference proceeds in two stages” (the amortised phase, followed by the iterative inference phase). The authors should address whether or not this two-phase design falls under the umbrella of “biologically plausible”. If it does, they should elaborate on how.

My remaining comments are more minor:

I don’t see any reference to figure 1 in the body text.

line 174 says “Equation 3 introduces an additional joint distribution over hidden causes and sensory data p(z,x)”. However, it is not clear to me how Eq. (3) introduces p(z,x). I would suggest the authors give a little more guidance.

line 179 says “minimising free energy with respect to theta will maximise p_theta (x) (see the second line of Equation 3)”. But p(x) is not in the second line. Do the authors mean the first line?

line 280: If I recall correctly, MINST has a total of 60,000 samples, which is usually split into 50,000 training samples, and 10,000 test samples.

line 296 refers to “weight normalisation”. What, precisely, is meant by that? And is it biologcially plausible?

line 299 says that the ADAM optimiser was used. That seems to contradict Algorithm 1. And how does this fit with the biological aspects of the model? A sentence or two on that would be helpful.

I really like the observation that the entropy of the classification nodes decreases during iterative inference. Question: Line 298 states that the top layer does not use an activation function. So, its values need not conform to a distribution. In fact, there could be values that fall outside the [0,1] range. How do they calculate the entropy of something which likely is not a valid distribution? This should be clarified.

In addition, I am fascinated by their experiments on the number of iterations it takes to arrive at a confident inference, and what that says about the difference between a task you’re learning, versus a task that you’ve practised and are expert on. Their allusion to the “Thinking Fast and Slow” book (by Daniel Kahneman) is apt. Given that their title is a play on Kahneman’s title, perhaps it’s appropriate to cite the book.

The results of the paper, and their relevance to cognitive neuroscience, are profound. I believe this paper should be published once the remaining wrinkles are ironed out.

Reviewer #2: In the current manuscript, the authors report the results of simulation experiments comparing performance of the ‘standard’ predictive coding (PC) model with their hybrid predictive coding (HPC) model using the MNIST data set. The HPC combines the generative component of the standard PC model with an amortized component. The amortized component provides a fast feedforward inference, which is subsequently refined by the generative component via the standard top-down processing of PC models. The parameters of the amortized component are then updated based on the final inference reached by the generative component.

The authors show that their HPC model has a number of attractive characteristics. For example, they show that it is more robust compared to the PC model when trained with small numbers of examples (and they provide some indication of why their model’s performance is superior in this context). Another attractive feature of their model is the fact that the balance between the contribution of the amortized vs. the generative component to the final inference falls directly out of their model architecture. Based on the results of their simulations, the authors claim that their model provides a neurophysiolgically plausible architecture that explains many previous findings that are challenging for standard PC models.

Overall, I like the paper. It addresses an interesting topic and is well written. It is straightforward and simple in the most positive way, which is not always the case with recent PC papers that can become unnecessarily complicated and convoluted. I think the paper is certainly worth being published in PLoS Comp Biol. However, I have a few comments that the authors might want to consider.

General comments (in no specific order):

1. While I agree that the model might capture some findings regarding visual processing, there are other aspects that I think are more difficult to map to biological vision. Specifically, if we take the results at face value and use HPC as a model of biological vision, we would have to conclude that we exclusively rely on feedforward processing with familiar inputs. While there is some evidence that feedback processing is less pronounced for familiar inputs, the current evidence suggests that exclusive reliance on feedforward processing is extremely unlikely to ever happen in human vision. I think this aspect needs to be addressed and discussed in more detail.

2. I don’t understand how the model helps to explain visual phenomenology or how it can speak to that topic. That gist perception is based on different processes than later focussed perception is well established. One might want to claim that these different processes ‘explain’ different visual phenomenology. The amount of explaining that’s being done is limited in my view: all that we can say is that there are different processes and that there might be different phenomenologies. Exactly how the processes lead to different phenomenologies – i.e., what the mechanisms are – remains totally unclear. But even if one accepts that the different processes of gist perception and focussed perception ‘explain’ phenomenology, I don’t understand what the model adds.

3. Related to my two previous points: prominent theories link phenomenology to recurrent processing (as far as I understand – I’m not an expert in consciousness research). Yet, the HPC model suggests that recurrent/feedback processing diminishes as we become more familiar with a stimulus set up to the point where vision is entirely based on feedforward processing. Thus, there seems to be a discrepancy between claims in the literature about feedback processing and phenomenology, and the claims made in this paper. According to HPC, if feedback processing determines phenomenology, then we should see a change in phenomenology as we become more familiar with a stimulus.

4. I would tone down the claims regarding the neurophysiological plausibility of the model (throughout the whole paper). I’m not convinced that the model is more (or less) neurophysiologically plausible than many other models that have tried to address a similar issue.

5. I would tone down some of the claims in the discussion/conclusion section. For instance, you say: ‘Hybrid predictive coding offers a new perspective on the biological relevance of the feedforward sweeps and iterative recurrent activity…’ I don’t think that’s true. There are many proposals – both conceptual and empirical – that have made similar suggestions before. In fact, in other parts of the paper, you use these other proposals to claim that ‘This account of perception is remarkably consistent with our proposed model.’ I think it would be more accurate to say that you provide a new model that captures old perspectives.

6. Page 8: You say that ‘Equation 3 introduces an additional joint distribution over hidden causes and sensory data p(z, x)…’. However, Equation 3 doesn’t introduce that joint distribution. It’s therefore not clear what you are referring to, which is confusing for the reader (or at least it was for me).

7. Page 14: More detail about the simulations are needed. Specifically, how do you assess accuracy of the amortized component over batches? As far as I understand what you do is this: 1. perform the amortized initial best guess and log that best guess as this component’s accuracy for the current input; 2. Refine the guess via the top-down component of the model and use the refined inference to update the amortized parameters. 3. Use the updated amortized parameters for the next input, and start with 1. Is that correct? If so, I think it would be worth spelling this out explicitly. In any case, in my view, it would be beneficial to be more explicit about exactly what you have done here.

8. Fig. 3 A: It’s worth highlighting and explaining why the standard PC model initially performs stronger than the HPC model but seems to show reduction in performance with increasing batch numbers.

9. Fig. 5 features several aspects that are a bit confusing (for my brain at least) but that are not highlighted or discussed: 1. The amortized component of the HPC model seems to perform better with fewer iterations. For instance, after 600 batches, it’s accuracy is at around 0.6 with 10 iterations but at around 0.5 with 100 iterations. Why does this happen? I find it difficult to get my head around it. 2. The HPC model seems to perform equally well independently of the number of iterations. How is that possible given that the generative component on its own struggles with the lower number of iterations? I assume that the amortized component compensates the difficulties of the generative component (which you kind of allude to). However, relating to my previous point, I find it difficult to see how the amortized component is able to perform better on low iterations (in particular, given that it’s updating is dependent on the accuracy of the generative component). 3. The standard PC shows interesting behavior with 25 and 50 iterations. It might be useful to highlight and explain both the better performance compared to HPC early on and the breakdown after more batches (also see my previous point).

10. Page 19: I’m not quite sure I understand why the learning with limited data isn’t shown for the classic PC model.

Minor comments:

1. I think you might want to reference Moshe Bar’s work more explicitly in the introduction. His proposal is one of the first that is very similar to what you suggest here in terms of the role of feedforward and feedback processing.

2. Page 4: I don’t quite understand this sentence: ‘Although classical bottom-up perspectives are often contrasted with Bayesian top-down theories in this debate, more nuanced pictures have also been proposed in which bottom-up and top-down signals both contribute to perceptual content, but in distinct ways.’ The original predictive coding (and similar) accounts also suggest that both bottom-up and top-down signals contribute to perceptual content. I think it would be useful to be more specific about what the cited proposals add.

3. Page 4: Related to the previous point, it might be useful to be more specific here: ‘Here, we develop, and illustrate with simulations, a novel computational architecture in which top-down and bottom-up signalling is adaptively combined to bring about perceptual representations within an extended predictive coding paradigm.’ As I said, bottom-up and top-down signals are combined in most accounts of perceptual processing. What’s special about your account? I think the key novelty of your proposal is the idea that predictions are not only implemented in top-down processing but integrated into bottom-up processing (as conceptually suggested by Teufel & Fletcher 2020).

4. Page 4: ‘In this setting, bottom-up signals convey prediction errors - the difference between predictions and data - whereas top-down signals convey the predictions.’ It might be worth highlighting that not all PC schemes work this way (e.g., Michael Spratling’s work).

5. Page 7: ‘…first to combine of iterative and amortized inference…’ Get rid of the ‘of’

6. Page 8: ‘…bound on Eq. 2, i.e. a quantity which…’ a comma is missing after ‘i.e.’

7. Page 8: ‘…parameters θ, e.g pθ(z, x).’ comma and full stop are missing after ‘e.g’. In fact, the missing comma after ‘i.e.’ and e.g.’ is a more general mistake throughout the paper.

8. Page 10: ‘…here denoted μ∗…’ I would say: ‘… denoted μ∗ in considerations of HPC below…’.

9. Page 10: This is a bit confusing ‘…going from data (stimuli) to hidden states (features)…’ as you previously used the term ‘features’ akin to ‘data’ when you say that predictive coding predicts features from objects.

10. Page 13: I would use ‘Learning of Amortized Parameters’ rather than just ‘Learning’ to avoid ambiguity in the illustration.

11. Page 13: I would suggest you start line 252 with something like: ‘Once inference is completed for the current input, generative and amortized parameters are updated via learning.’

12. Page 19: ‘…incorporating a bottom, amortised component…’ It should be ‘bottom-up’ rather than ‘bottom’

13. Page 19: ‘Note THAT we still use the complete test set of 10,000 images for testing.’

14. Page 19: ‘To gain intuition further intuition…’ get rid of the first ‘intuition’

15. Page 21: Fig. 7B: I assume you used a stopping criterion here (i.e., the number of iterations required to reach a specific criterion)? I might have missed it but I think that might not be specified anywhere.

Reviewer #3: Review is uploaded as an attachment.

**Have the authors made all data and (if applicable) computational code underlying the findings in their manuscript fully available?**

Reviewer #1: **No: **I'm not aware of any code being made available.

Reviewer #2: Yes

Reviewer #3: Yes

PLOS authors have the option to publish the peer review history of their article (what does this mean?). If published, this will include your full peer review and any attached files.

Reviewer #1: No

Reviewer #2: No

Reviewer #3: No
---

## [Decision Letter · Decision Letter 1]

22 Apr 2023

Dear Dr. Millidge,

Thank you very much for submitting your manuscript "Hybrid Predictive Coding: Inferring, Fast and Slow" for consideration at PLOS Computational Biology. As with all papers reviewed by the journal, your manuscript was reviewed by members of the editorial board and by several independent reviewers. The reviewers appreciated the attention to an important topic. Based on the reviews, we are likely to accept this manuscript for publication, providing that you modify the manuscript according to the review recommendations.

As you will se the three reviewers still have some concerns, especially with regard to discussion and interpretation. While any changes are not likely to be extensive (i.e. minor) they are nevertheless important for the publication.

Sincerely,

Ulrik R. Beierholm

Academic Editor

PLOS Computational Biology

Marieke van Vugt

Section Editor

PLOS Computational Biology

Reviewer's Responses to Questions

**Comments to the Authors:**

Reviewer #1: The authors have done a good job of addressing most of my concerns from the first review. I do have a few relatively minor questions and comments.

The biggest question still in my mind is that of biological plausibility. The biological plausibility (BP) of their method is a major theme of the paper, and is stated throughout, including the Abstract. For example, the second bullet point in the Highlights section states,

“Our hybrid predictive coding network jointly optimizes a single unified energy function by combining both iterative and amortized inference and is trained with biologically plausible prediction error minimization using only local Hebbian updates.”

My main question, in this vein, is about how the amortized network is trained. According to [Bogacz, 2017], which is cited in the footnote in the 1st paragraph of section 2.3, biological plausibility adheres to two main criteria: (1) local computation, and (2) local plasticity. Local plasticity is described (in that paper) as the restriction that “Synaptic plasticity is only based on the activity of pre-synaptic and post-synaptic neurons.” I agree that the predictive-coding part of HPC satisfies those BP criteria. However, it’s not clear to me how the connection weights in the amortized network could be learned using a local plasticity (Hebbian) update rule. Since the authors are somewhat vague about how the corresponding error nodes are situated in their network (they excluded them from fig. 2), it is difficult to assess the validity of the BP claim.

The BP of weight normalization is another aspect of the HPC method that is unclear. In line 313, the authors state that they keep the mean of the amortization weights constant, but they don’t say how. I raised this question in my first review, and the authors responded with “[weight normalization] may or may not be biologically plausible…”, in addition to a reference to an arXiv paper. I think it would be prudent to specify which type of normalization they actually used. Not specifying, but making claims of BP throughout the paper suggests that they implemented weight normalization in a BP manner, which might be inaccurate, or even misleading.

On line 434, the authors mention “normalizing the values for the final layer to obtain an output probability distribution”. Again, I would encourage the authors to be more explicit regarding the degree to which their implementation is BP. Moreover, what kind of normalization was used? Was it softmax, or some other type of scaling?

It is fine if parts of the algorithm are less biologically realistic than other parts. My main concern is about being transparent and precise about claims of BP. I would suggest the authors either describe in fuller detail how the amortized learning rule and normalizations are BP, or simply back away from those claims.

On line 358, they state “The accuracy of amortised inference increases at a slower rate, consistent with the intuition that learning discriminative models requires more data relative to generative models.” However, the slower training of the amortized model could also be explained by the fact that it cannot learn until the PC network learns, since the amortized model is really just learning to predict the equilibrium state of the PC model. To me, this seems like a much more likely (and prosaic) explanation.

On line 386, the authors refer to results that have not yet been presented; figure 5 seems to have the relevant data, but that figure is not referred to until the next paragraph. It seems that the paragraphs might have been swapped. Also, “iterative iterations” seems redundant. Perhaps they mean “inference iterations”.

With regard to MNIST, in my first review I questioned how many samples were in the training set, and said that I thought there were only 50,000. Looking into it further, it seems that the authors might have been correct in their first draft. I’m not too concerned with this particular detail, but just wanted to come clean on the issue. I trust the authors will know, from their own experiments, how many samples they trained on, and report that in the paper.

A natural, but unaddressed question is whether the HPC network converges to the same state every time. Does the initialization by the amortized network cause the iterative inference of the PC network to settle into a different final state than it would have if it was initialized differently? If so, under what circumstances?

Other comments:

- Line 391: The statement starting “Finally, the instabilities…” is not a sentence.

- Equation (10): Should there be a prior probability factor included in the product, p(z_L)?

- Line 164: The authors are comparing HPC to SPE. In the sentence, “In terms of technical content, the model’s bidirectional connectivity is between individual neurons, rather than between layers”. The authors should clarify that they are referring to the SPE model (I believe). Also, I don’t think it’s fair to say that the SPE model’s connectivity is between neurons, and not between layers. Or, at least, it’s not clear what is meant by that.

- References: There are some typographical errors in the References section. In particular, proper nouns should be capitalized, like “Helmholtz”, “Bayesian”, “Hebbian”, and “Kalman”, as well as acronyms, like “VAE”, and “FMRI”. Also, I believe the arXiv paper [Millidge, Tschantz, Seth, Buckley, 2021] has been published, and that published version should be cited instead. Perhaps there are others.

In summary, this is an excellent paper that should be published once these rather minor wrinkles are ironed out.

Reviewer #2: The authors’ revision of their paper in response to my comments is sparse. Personally, I think the authors have missed an opportunity to increase the appeal of their paper to a wider audience by engaging with the comments in a bit more detail. But overall, they have addressed most of my comments satisfactorily.

The only remaining issue I have is the discussion of the link between the model and phenomenology. In my view, this point remains relatively weak. I would urge them to consider being more specific and detailed in their discussion of this issue:

Comment 2 (in the authors' responses)

I think the authors either need to unpack the issue of how their model relates to phenomenology in far more detail – even if this point is speculative – , or remove it. If they wish to retain it, the authors need to explain in detail how the ‘different correlates provide plausible accounts for their corresponding phenomenological properties.’ In other words, which correlates correspond to which phenomenological properties, and how do they provide plausible accounts?

Comment 3 (in the authors' responses):

I don’t think that the authors have responded to my comment, which might be my fault for not being clear enough. My point is that there is a tension between those theories of consciousness that emphasise recurrent processing as a key mechanism and the authors’ HPC model. In my view, if the authors wish to retain the comments linking HPC to phenomenology, then they need to detail how their account relates to theories of consciousness that emphasise recurrent processing. This is important because, as far as I understand, the authors claim that the specific characteristic of conscious phenomenology is determined by the balance between the amortized and the iterative components, while theories of recurrent processing emphasis the recurrent component exclusively.

Reviewer #3: My review is uploaded as an attachment.

**Have the authors made all data and (if applicable) computational code underlying the findings in their manuscript fully available?**

Reviewer #1: Yes

Reviewer #2: Yes

Reviewer #3: Yes

PLOS authors have the option to publish the peer review history of their article (what does this mean?). If published, this will include your full peer review and any attached files.

Reviewer #1: No

Reviewer #2: No

Reviewer #3: No

Figure Files:

Data Requirements:

Reproducibility:

References:

---

## [Editor Report · Decision Letter 2]

20 Jun 2023

Dear Dr. Millidge,

We are pleased to inform you that your manuscript 'Hybrid Predictive Coding: Inferring, Fast and Slow' has been provisionally accepted for publication in PLOS Computational Biology.

Best regards,

Ulrik R. Beierholm

Academic Editor

PLOS Computational Biology

Marieke van Vugt

Section Editor

PLOS Computational Biology

---

## [Editor Report · Acceptance letter]

18 Jul 2023

PCOMPBIOL-D-22-01276R2 

Hybrid Predictive Coding: Inferring, Fast and Slow

Dear Dr Millidge,

I am pleased to inform you that your manuscript has been formally accepted for publication in PLOS Computational Biology. Your manuscript is now with our production department and you will be notified of the publication date in due course.

With kind regards,

Zsofia Freund
